# Young-Onset Dementia: Clinical Findings and Factors That Delay Early Diagnosis—A Retrospective Observational Study

**DOI:** 10.3390/biomedicines13112793

**Published:** 2025-11-17

**Authors:** Juan Rivas, Mauricio Hernández, Jose Miguel Erazo, Oscar Arango, Paulina Cortés, Jennifer Lasso, Simon Giraldo, Carlos Miranda

**Affiliations:** 1Hospital Departamental Psiquiátrico Universitario del Valle, Cali 760034, Colombia; jose.miguel.erazo@correounivalle.edu.co (J.M.E.); lasso.jennifer@correounivalle.edu.co (J.L.); carlos.miranda@correounivalle.edu.co (C.M.); 2Departamento de Psiquiatría, Universidad del Valle, Cali 760043, Colombia; mauricio.hernandez@correounivalle.edu.co (M.H.); oscar.roldan@correounivalle.edu.co (O.A.); maria.paulina.cortes@correounivalle.edu.co (P.C.); simongiraldooec@gmail.com (S.G.); 3Departamento de Psiquiatría, Fundación Valle del Lili, Cali 760035, Colombia; 4Facultad de Ciencias de la Salud, Universidad Icesi, Cali 760001, Colombia

**Keywords:** early-onset dementia, delayed diagnosis, preventable risk factors, high blood pressure, diabetes, irritability

## Abstract

**Background/Objectives**: Young-onset dementia (YOD) is a form of dementia where symptoms appear before the age of 65 years with a worse course, a poorer prognosis, and a lower survival rate than late-onset dementia. Psychiatric disorders often entail confusion, which delays their diagnosis and management. This study emphasizes the risk factors and confounders that limit opportunities to provide adequate early diagnoses of YOD. **Methods**: A retrospective, analytical, and observational study was based on the clinical records of 191 patients with a diagnosis of probable YOD in a medium-complexity hospital between 2009 and 2024. Demographic variables and the characteristics of the population were analyzed. An explanatory linear regression analysis was conducted to highlight the time required for diagnosis beginning at the onset of symptoms. **Results**: A high proportion of initial misdiagnoses were identified, and most patients were initially diagnosed with psychiatric or neurological disorders other than dementia. The main preventable risk factors were high blood pressure (HBP), type 2 diabetes mellitus (T2DM), and traumatic brain injury (TBI). HBP and the presence of irritability were associated with earlier diagnosis, whereas T2DM and the initial diagnosis of an affective or anxiety disorder were associated with a longer delay prior to diagnosis. **Conclusions**: Due to delays in seeking care and initial misdiagnoses as affective or anxiety disorders, T2DM is associated with a delayed final dementia diagnosis. In contrast, HBP and irritability were linked to shorter diagnostic times. These findings underscore the need for improved diagnostic capacity, adapted clinical tools, and awareness strategies to promote the early recognition of YOD.

## 1. Introduction

Dementia is a clinical syndrome characterized by a progressive decline in cognitive functions and the loss of autonomy in daily activities. Its impact extends beyond the individual, affecting families, caregivers, and healthcare systems [1]. Globally, approximately 50 million people live with dementia, and this number is projected to rise to 152 million by 2050, with two-thirds of cases occurring in low- and middle-income countries [2].

Young-onset dementia (YOD) is a form of dementia that affects younger people, with symptoms beginning before the age of 65. Although advanced age is the strongest known risk factor in dementia, YOD is not an inevitable consequence of biological aging [3]. The estimated global prevalence is 119 cases per 100,000 people aged 30–64, representing approximately 3.9 million individuals [4]. Compared to late-onset dementia (LOD), YOD is associated with faster progression and shorter survival [5]. The most common etiologies include early-onset Alzheimer’s disease (EOAD), frontotemporal dementia (FTD), and dementia with Lewy bodies (LBD) [3,6].

In Colombia, approximately 800,000 people are estimated to live with dementia, with Alzheimer’s disease (AD) being the most frequent cause [7]. YOD represents a particular challenge in diagnosis and management, especially in Valle del Cauca and the city of Cali, where an increase in reported AD cases has been observed in recent years [8].

The diagnosis of YOD is complex. It could be explained by its heterogeneous presentation, which may include language disturbances, visuospatial impairment, behavioral and personality changes, affective or psychotic symptoms, apathy, or anxiety [9,10]. Another important factor is the limited awareness and scarce research on this condition, which contributes to delays in diagnosis [11,12]. These delays are further influenced by systemic barriers, stigma, and diagnostic challenges, including the tendency of both professionals and families to attribute symptoms to psychiatric or situational causes [11,12].

No disease-modifying therapies are currently available for YOD. Given its rapid course and poor prognosis, research into symptomatic and non-pharmacological interventions remains important. Techniques such as transcranial magnetic stimulation have been investigated as potential approaches to enhance synaptic plasticity and cognition, but the evidence is still preliminary and requires further validation [13,14,15,16].

Despite advances in our understanding of neurocognitive disorders, a gap persists in the scientific literature regarding the factors that influence diagnosis in this specific context. Early recognition is essential to provide timely clinical management, improve patients’ quality of life, and support caregivers. However, most studies have been conducted in Europe, North America, and Asia, with few focusing on Latin America. This underscores the urgent need for epidemiological and clinical research in this region.

## 2. Materials and Methods

### 2.1. Study Design

This is a retrospective observational cohort study with an analytical component. Clinical records of patients with probable young-onset dementia (YOD), who were evaluated at the Hospital Departamental Psiquiátrico Universitario del Valle between 2009 and 2024, were reviewed.

### 2.2. Population and Sample

A total of 204 clinical records of patients with a diagnosis of probable YOD were initially reviewed. Thirteen records (6.4%) were excluded due to missing data greater than 10%, leaving 191 patients for the final analysis. Inclusion criteria were patients older than 18 years whose symptom onset occurred before the age of 65 and who received care at the Hospital Departamental Psiquiátrico Universitario del Valle between 2009 and 2024. Exclusion criteria were incomplete records, insufficient information to confirm the diagnosis, or lack of adequate data regarding the clinical course and relevant study variables.

Diagnoses of EOAD, vascular dementia (VaD), mixed dementia (MD), that is, VaD plus AD, behavioral variant frontotemporal dementia (bvFTD), and progressive supranuclear palsy (PSP) were observed. Patients with Parkinson’s disease or LBD were not included because they are managed by another institution. The patients were diagnosed in the following manner. Two psychiatrists who were experts in the field of neuropsychiatry reviewed the quality of the data contained in the medical records to determine the presence of cognitive impairment. All patients were assessed following the evaluation protocol for patients with suspected dementia, developed at the hospital in accordance with international guidelines. This includes brain magnetic resonance imaging (MRI) to identify the findings that were suggestive of each pathology. Subsequently, standardized neuropsychological tests were performed. Once cognitive deterioration was confirmed in this manner; if it was indicated, the patients underwent a positron emission tomography (PET) scan with 18-fluorodexoglucose in cases of diagnostic uncertainty: 41 patients with EOAD; 26 with FTD; 2 with VaD; and 1 with PSP. On the basis of this information, the experts in neuropsychiatry classified the probable etiology of the case of dementia in question. Inter-rater reliability (kappa) was 0.8. Discrepancies between evaluators were resolved by a psychiatrist with a fellowship in dementia. Initially, the International Statistical Classification of Diseases and Related Health Problems-10 (ICD-10) [17] criteria were used in all patients, followed by the specific criteria for each pathology, with the aim of obtaining a more accurate diagnosis in accordance with international criteria. We used ICD-10 because it is the approved system of registry by the Health Ministry in Colombia. The diagnoses were made based on the review of the medical records, and the criteria for each disorder were applied at the end of the analysis. For each specific condition, the current diagnostic criteria were used. Biomarkers were not used because they are not available in our setting and, therefore, have not been included in our hospital’s evaluation protocols.

For EOAD and MD, the criteria proposed by the National Institute of Neurological and Communicative Disorders and Stroke, Alzheimer’s Disease and Related Disorders Association (NINCDS-ADRDA) were used [18]. For VaD, the criteria suggested by the National Institute of Neurological Disorders and Stroke (NINDS) and the Association Internationale pour la Recherche et l’Enseignement en Neurosciences (AIREN) were used [19]. For FTD, criteria proposed by the international consortium were employed [20]. For PSP, the criteria suggested by the Movement Disorder Society were used [21]. For the applications of these diagnostic criteria, the latest versions of each were used and the process was carried out by two experts in neuropsychiatry.

### 2.3. Variables

Sociodemographic variables were collected for both groups and subgroups, including sex; age at the onset of symptoms; age at the first consultation; age at the final diagnosis; marital status; type of caregiver; sex of caregiver; comorbidities; both initial and final symptoms, particularly depression, anxiety and alterations in memory, sleep, irritability, motor skills, language, personal care, nutrition, isolation, and psychosis; and the diagnosis of dementia according to the International Statistical Classification of Diseases and Related Health Problems-10 (ICD-10).

### 2.4. Statistical Analysis

Qualitative demographic characteristics, such as sex, marital status, type of caregiver, and caregiver sex were analyzed in terms of absolute and relative frequencies. Quantitative variables, including age at the onset of symptoms, age at the first consultation and the intervals (years) from symptom onset to first consultation, from first consultation to diagnosis, and from symptom onset to diagnosis, were summarized using measures of central tendency and dispersion. Variables with approximately normal distributions were reported as mean ± standard deviation (SD), whereas skewed variables were reported as median and interquartile range (IQR).

The analytical component lies in the comparison of demographic and clinical characteristics across diagnostic subtypes and in the use of regression models to explore factors associated with diagnostic delay. The study followed the STROBE recommendations for reporting observational research. Strobe diagram and checklist are provided as an annex.

Comorbidities, baseline symptoms, and symptoms at diagnosis were compared across diagnostic groups using relative frequency analysis. Similarly, age and diagnostic intervals were examined in relation to the type of dementia, applying the same descriptive approach (mean ± SD or median [IQR], depending on distribution). Finally, an explanatory linear regression analysis was conducted to explore the factors associated with diagnostic delay from symptom onset, including all demographic and clinical variables under investigation as regressors. Statistical analyses were performed using IBM SPSS Statistics for Windows, version 26.0 (IBM Corp., Armonk, NY, USA).

### 2.5. Ethical Considerations

Given that this is a retrospective study based on the review of patients’ medical records, the requirement for obtaining explicit informed consent did not apply. The study is conducted under the scope of the general informed consent signed by patients upon hospital admission, which expressly authorizes the use of clinical information pertaining to their condition, provided that such information is handled in an anonymized manner and under strict standards of confidentiality. The confidentiality and anonymity of the information were guaranteed in accordance with relevant national and international health research regulations. The Internal Review Board approved the protocol (code 032-2024).

## 3. Results

A total of 191 patients with a diagnosis of YOD were included in this research. In the sample, 63.4% of participants were women, 53.9% had a stable partner, and in 70.7% of cases, the primary caregiver was female. The participants’ diagnoses were as follows: 53.4% of participants were diagnosed with EOAD, 19.4% with VaD, 18.3% with an FTD, 5.2% with a PSP, and 3.7% with MD.

The initial diagnosis refers to the diagnosis made at the first contact with the patient. Since the evaluation at that time was purely clinical, it was not possible to establish a probable diagnosis. In 58.6% of cases, the initial diagnosis was unspecified dementia, whereas, in the remaining 41.4% of cases, the possibility of dementia was not considered. Among these patients, 11.5% were treated for an affective disorder, 9.4% for an anxiety disorder, 7.3% for schizophrenia, 3.1% for mild cognitive impairment, 1.6% for Parkinson’s disease (PD), and 8.4% for other psychiatric diagnoses.

In cases in which the initial diagnosis was an affective or anxiety disorder, the patients were diagnosed with FTD in 17.1% and 11.4% of the cases, respectively, followed by VaD in 13.4% and 10.7% of cases, respectively, and EOAD in 10.7% and 8.8% of the cases, respectively. On the other hand, when patients were diagnosed with psychosis or schizophrenia, 42.6% received a probable diagnosis of VaD. In cases in which the initial diagnosis was PD, the final diagnosis highlighted probable PSP in 29.8% of the patients. When these diagnoses were compared, the chi-square test led to the rejection of the null hypothesis of independence between the initial and final diagnoses, thus indicating a statistically significant association between the initial and final diagnoses (*p* < 0.05) (Table 1).

Among the preventable risk factors, high blood pressure (HBP) was the most common pathology (35.4%). Similarly, type 2 diabetes mellitus (T2DM) occurred in 14% of patients, a history of traumatic brain injury (TBI) occurred in 11.2%, and dyslipidemia occurred in 10.5%. HBP was especially prevalent among patients in the MD and VaD groups (71.4% and 62.2%, respectively) (Table 2).

With respect to initial symptoms, memory impairment was identified in 37.9% of patients, predominantly in 68.3% of patients with EOAD. Motor symptoms were observed in 80% of patients in the PSP group and in 18.9% of patients in the VaD group. Affective symptoms such as depression and anxiety were frequent among patients in all subgroups, with a higher prevalence of depression in VaD patients (44.1%) and a higher prevalence of anxiety in FTD patients (31.4%) (Table 3).

With respect to the evolution of symptoms, strong impacts on instrumental activities in daily life (92.8%) and memory (90.0%) were observed. The most frequently observed behavioral symptoms were anxiety (80.3%), irritability/lability (75.9%) and depressive/dysphoric symptoms (67.8%). Motor symptoms persisted as a predominant finding in 100% of PSP patients and, to a lesser extent, in 45.9% of VaD patients (Table 4).

In general, the median age of onset of patients’ symptoms was 57 years, whereas the age at diagnosis was 60 years. The median time from the onset of symptoms to the first consultation was 2 years, and that from the onset of symptoms to diagnosis was 3 years. According to the final diagnosis, the median age of onset of symptoms was lower in patients with FTD (55 years), followed by those with EOAD and PSP. In contrast, MD and VaD were associated with later ages of onset at 60 and 58 years, respectively. With respect to patients’ age at the time of the first consultation, a similar trend was observed, such that FTD and EOAD patients exhibited lower ages (57 years), whereas the corresponding value for MD and VaD patients was 61 years. Minor differences were observed in patients’ age at diagnosis, with medians ranging between 59 and 62 years; FTD was once again associated with the earliest age at 60 years.

The time from the onset of symptoms to the first visit was longer among patients in the FTD group (median: 2 years; IQR: 3) and shorter among those in the MD and VaD groups (1 year in both patients). The interval between the first consultation and the diagnosis was generally short, with medians of less than 1 year among patients in the MD and VaD groups and medians of 1 to 2 years among patients in the other groups. Finally, the total time from the onset of symptoms to diagnosis was significantly longer among patients in the FTD group (5 years; IQR: 4) and shorter among those in the MD group (1-year IQR: 6); this figure was approximately 3 years for patients in the other groups (Figure 1).

The time from the onset of symptoms to the final diagnosis was explained on the basis of a linear regression model, in which the time from the onset of symptoms to the first consultation, the presence of an initial diagnosis of affective disorders and/or anxiety disorders, the presence of comorbidities such as HBP and/or T2DM, and initial symptoms of irritability were included as predictors. These factors made statistically significant contributions to the model, according to an analysis of variance (ANOVA, Statistical analyses were performed using IBM SPSS Statistics for Windows, version 26.0 (IBM Corp., Armonk, NY, USA).), which led to the rejection of the null hypothesis that all regression coefficients are equal to zero (i.e., that they do not contribute to the model). In contrast, the variables were demonstrated to contribute to the explanation of the time prior to diagnosis, with a p value of 0.0001. In addition, the coefficient of determination (R^2^) was 0.601, thus indicating that the model correctly explained 60.1% of the total variation in the time prior to diagnosis.

In support of the assumptions of the model, the Durbin–Watson statistic was 1.943, thus suggesting the independence of the residuals. Similarly, the Shapiro–Wilk test pertaining to the errors was not significant (*p* > 0.05), thus indicating normality in the distribution of the residuals; accordingly, the assumptions of independence and normality required for the model were satisfied. On the other hand, the collinearity analysis indicated adequate independence among the predictors, with variance inflation factor (VIF) values ranging between 1.015 and 1.260, thus ruling out the possibility of relevant problems with multicollinearity.

Among the predictors included in this research, the time from the onset of symptoms to the first consultation exhibited the strongest and most significant association with the factor under investigation (B = 0.918; *p* < 0.001), thus indicating that, for each year between the onset of symptoms and the time when the patient attends a consultation, the time required for the final diagnosis increases by 0.918 years.

Similarly, patients who were initially diagnosed with affective disorders (B = 1.101; *p* = 0.023) and/or anxiety disorders (B = 1.389; *p* = 0.011) exhibited increases of 1.1 and 1.3 years, respectively, in the time before the final diagnosis. This finding implies that a patient who has been diagnosed with these conditions could take longer to receive a definitive diagnosis (Table 5).

Finally, the presence of irritability was also negatively and significantly associated (B = −0.897; *p* = 0.035), thus indicating that individuals who exhibited symptoms of irritability at their initial consultation were associated with a reduction of 0.89 years in the time prior to a definitive diagnosis (Table 5).

## 4. Discussion

In this cohort of 191 patients with YOD, women were predominant, representing 63.4% of the sample. This pattern is consistent with previous studies. A recent meta-analysis reported an age-standardized global prevalence of 119 cases per 100,000 people between 30 and 64 of age, including higher rates in women (159 per 100,00) than in men (114 per 100,000) [4].

The most frequent diagnosis was EAOD, accounting for 53.4% of cases, a finding that aligns with other clinical series [4]. Syndromes associated with FTD and tauopathies such as PSP accounted for a significant proportion of cases [22,23]. Although these subtypes are less prevalent, they have considerable functional impact and require a differentiated diagnostic approach.

### 4.1. Diagnostic Time

A central finding of this study was the high proportion of initial misdiagnoses observed in this context. Almost half the patients were first treated for different psychiatric or neurological conditions, and the patients were initially treated as if they had different psychiatric or neurological disorders. This diagnostic uncertainty has been described before in YOD, where clinical presentations are often atypical. Early symptoms may include affective, behavioral, or motor changes that appear before clear cognitive impairment. When the onset of the disease is very early and involve prominent behavioral symptoms, the delay reaching a probable diagnosis can ranges between 5 months and 18 years [24]. However, one study reported that, although patients with an early onset of symptoms seek consultations later, their final diagnosis is reached more quickly [25]. The percentage of misdiagnoses varies between 50% and 71.4%, and a high frequency of diagnoses of affective disorders, anxiety, and depression is observed in this context, although these disorders are rare at late ages [26,27]. The early presence of psychotic and affective symptoms often misleads clinicians, underscoring the need for specific diagnostic guidelines in YOD. The main modifiable risk factors identified in this context were HBP, T2DM, and TBI. These findings consistent with previous studies linking such conditions to an increased risk of dementia, particularly in vascular and mixed forms [28]. These findings are also consistent with studies such as those conducted by Javanshiri et al. [29] who reported strong associations among HBP, VaD, and MD, suggesting that vascular damage plays a shared pathogenic role in cognitive decline. The review conducted by Gottesman et al. [30] highlighted TBI in midlife as one of the most important modifiable risk factors for pre-venting dementia, including AD and MD. T2DM was present in 14% of cases, also significantly prevalent (14%), especially in patients with VaD and DM. This finding is in line with the results reported by Lennon et al. (2023), who reported that T2DM increases the risk of dementia, particularly when combined with the presence of other vascular factors [31].

TBI was identified in 11.2%, with the highest prevalence in those with PSP (30%). This finding is in line with recent evidence that has linked TBI with an increased risk of developing tauopathies, particularly among young people. Asken et al. reported that exposure to cranial impacts—particularly repeated impacts—is associated with earlier symptom onset in FTD, both in the behavioral variant and primary progressive aphasia. They also described a dose-dependent relationship between years of exposure and the age of onset of symptoms [32]. In addition, Garder et al. reported that moderate or severe TBI significantly increases the risk of dementia in comparison with other nonbrain injuries, especially among young adults [33].

Finally, the initial symptoms varied by diagnostic subtype. Memory impairment was the most frequent in YOD patients, while the motor symptoms predominated in PSP. Affective symptoms, including depression and anxiety, were common across all subgroups. These overlapping features can contribute to diagnostic confusion, especially in nonspecialized clinical settings.

### 4.2. Symptoms

In this cohort, significant functional impairment was evident from the early stages of the disease. A total of 92.8% of patients showed deterioration in the instrumental activities of daily living, and 77.55% showed deterioration in basic activities. This indicates a high degree of dependence, even during the moderate phases of the disorder. These findings are consistent with those reported by Makino et al., who reported that limitations in instrumental activities are associated with an increased risk of progression to dementia, particularly when they coexist with mild cognitive impairment [34].

The neuropsychiatric profile of the participants also revealed high rates of symptoms. Anxiety was present in 80.3% of patients, irritability, or emotional lability in 75.9%, and depressive or dysphoric symptoms in 67.8%. These results align with the findings of Ismail et al. (2018), who identified such symptoms as frequent prodromal markers of early-onset dementias. Within this context, they are grouped under the concept of “mild behavioral impairment” [35].

Motor disorders were another relevant clinical feature. They were observed in 100% of PSP patients and 45.9% of VaD patients, reinforcing the presence of a mixed degenera-tive-vascular component in this subgroup. This pattern has also been described by Ramli (2024), who documented the coexistence of motor symptoms and cognitive impairment in cases of MD with a PSP component [36].

### 4.3. Predictors of Diagnostic Delay

One of the most relevant contributions of this study lies in its identification of significant predictors of the time prior to diagnosis based on a linear regression model that exhibited a good fit (R^2^ = 0.601). In particular, consultation delays represented the most influential factor. This finding is consistent with recent research. For example, one study reported an average time to diagnosis of 3.4 years. Younger age at symptoms onset, dementia subtypes other than AD of bvFTD, and a higher number of services consulted were all associated with additional delays, ranging from days to months. In contrast, access to specialized YOD services shortened the time to diagnosis by nearly 12 months [37].

Similarly, an initial diagnosis of affective or anxiety disorder was linked to an average delay of as long as 2.4 years, a finding that is consistent with that the results presented by Ismail et al. [35], who noted that affective and emotional dysregulation symptoms may represent prodromal manifestations of dementia but are usually interpreted as primary psychiatric disorders, thus contributing to the diagnostic delay observed in this context.

Comorbidities also play an important role. T2DM was associated with a significant diagnostic delay, possibly due to the clinical overlap between cognitive impairment and metabolic dysfunction. Supporting these interpretations, Barbiellini Amidei et al. (2021) demonstrated that early-onset diabetes increases the risk of dementia and can mask initial symptoms, thus making this disorder difficult to recognize [38].

In contrast, HBP was marginally associated with a shorter time to diagnosis. This may be explained by the greater contact with healthcare services and more frequent clinical monitoring among these patients, as suggested by Shang et al. [39]. Finally, irritability was associated with a shorter diagnostic delay, possibly because of its disruptive nature and its frequent association with neurodegenerative syndromes such as FTD, in which behavioral symptoms often trigger earlier referral [35].

## 5. Conclusions

One of the most relevant contributions of this study lies in its identification of significant possible predictors of the time prior to diagnosis. The delay in consulting was the most influential predictor in this context, thus highlighting the importance of interventions that promote the early recognition of symptoms by patients, families, and general practitioners. This finding also suggests the need to develop strategies within the health system that can offer these patients easier access to specialized personnel. Also, an initial diagnosis of affective or anxiety disorders was related to a significant delay, thus highlighting the need to improve the ability of relevant actors in the clinical setting to differentiate between primary psychiatric symptoms and early neurodegenerative manifestations. T2DM was related to a diagnostic delay, possibly as a result of overlapping symptoms or the attribution of cognitive deficits to metabolic causes. In contrast, hypertension was marginally related with a shorter time prior to diagnosis, which could be explained by a longer period of medical follow-up for these patients. Finally, the presence of irritability was related to a shorter time until diagnosis, which could be due to the fact that this symptom is more notable or disruptive for caregivers and professionals, thus motivating a faster diagnostic search.

This study emphasizes the need to strengthen diagnostic capacities at both the primary and specialized levels of care. Also, it is necessary to follow clinical protocols that have been adapted to the task of identifying YOD. Finally, the findings of this research highlight the importance of a multidisciplinary approach that can take behavioral symptoms, functional impact, and medical comorbidities into account from a comprehensive perspective.

### Limitations

The sample size represents an important limitation of this study. However, the frequency of YOD is low, and although the findings of this research cannot be extrapolated to the general population, attention should be given to the causes and consequences of a late diagnosis in this context.

Furthermore, the sample only includes patients who presented to the Hospital Departamental Psiquiátrico Universitario del Valle with a diagnosis of YOD, which represents a possible selection bias; it is possible that additional patients remain misdiagnosed or are not correctly coded in clinical records. This also suggests that the data indicate potential gaps in clinical awareness or challenges in differentiating YOD from primary psychiatric disorders in its early stages. Finally, it should be noted that the data used have inherent limitations, as they rely on the accuracy and completeness of medical record documentation.

## Figures and Tables

**Figure 1 biomedicines-13-02793-f001:**
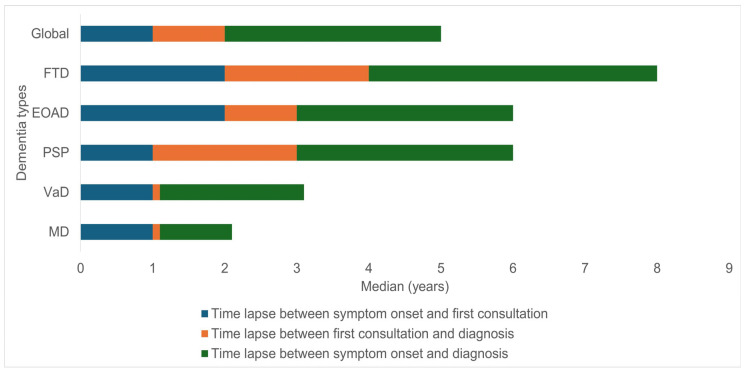
Details of patients’ age and time at first consultation and diagnosis according to the type of YOD (*n* = 191) (Hospital Departamental Psiquiátrico Universitario del Valle during the period 2009–2024). EOAD: early-onset Alzheimer’s disease. MD: mixed dementia. FTD: frontotemporal dementia. PSP: progressive supranuclear palsy, VaD: vascular dementia.

**Table 1 biomedicines-13-02793-t001:** Initial diagnosis at the first consultation vs. final diagnosis of YOD (*n* = 191) (Hospital Departamental Psiquiátrico Universitario del Valle during the period 2009–2024).

Initial Diagnosis	Final Diagnosis					*p* Value Chi ^2^ *
	EOAD (*n* = 102) *n* (%)	MD (*n* = 7) *n* (%)	FTD (*n* = 35) *n* (%)	PSP (*n* = 10) *n* (%)	VaD (*n* = 37) *n* (%)	
Dementia (*n* = 112)	67 (65.7)	3 (43.0)	19 (54.3)	5 (50.0)	18 (48.7)	0.0001
Bipolar disorder (*n* = 22)	11 (10.8)	0 (0.0)	6 (17.1)	0 (0.0)	5 (13.5)	
Anxiety disorder (*n* = 18)	9 (8.8)	0 (0.0)	4 (11.4)	1 (10.0)	4 (10.8)	
Other (*n* = 16)	5 (4.9)	1 (14.3)	2 (5.7)	1 (10.0)	7 (18.9)	
Schizophrenia (*n* = 14)	6 (5.9)	3 (42.6)	2 (5.7)	0 (0.0)	3 (8.1)	
Mild cognitive impairment (*n* = 6)	4 (3.9)	0 (0.0)	2 (5.7)	0 (0.0)	0 (0.0)	
Parkinson disease (*n* = 3)	0 (0.0)	0 (0.0)	0 (0.0)	3 (30.0)	0 (0.0)	

EOAD: early-onset Alzheimer’s disease. MD: mixed dementia. FTD: frontotemporal dementia. PSP: progressive supranuclear palsy, VaD: vascular dementia. * *p* Value Chi^2^: test comparing initial diagnoses across final diagnosis groups.

**Table 2 biomedicines-13-02793-t002:** Comorbidities according to the type of YOD (*n* = 191) (Hospital Departamental Psiquiátrico Universitario del Valle during the period 2009–2024).

Diagnosis	EOAD (*n* = 102)	MD (*n* = 7)	FTD (*n* = 35)	PSP (*n* = 10)	VaD (*n* = 37)	Global (*n* = 191, %)
	*n* (%)	*n* (%)	*n* (%)	*n* (%)	*n* (%)	
High blood pressure (*n* = 48)	11 (10.8)	5 (71.4)	8 (22.9)	1 (10.0)	23 (62.2)	(25.1)
Hypothyroidism (*n* = 21)	10 (9.8)	0 (0.0)	3 (8.6)	0 (0.0)	8 (21.6)	(11.0)
Traumatic brain injury (*n* = 17)	7 (6.9)	0 (0.0)	2 (5.7)	3 (30.0)	5 (13.5)	(8.9)
Type 2 Diabetes mellitus (*n* = 15)	1 (1.0)	2 (28.6)	5 (14.3)	1 (10.0)	6 (16.2)	(7.9)
Dyslipidemia (*n* = 12)	6 (5.9)	2 (28.6)	1 (2.9)	1 (10.0)	2 (5.4)	(6.3)
Epilepsy (*n* = 12)	6 (5.9)	0 (0.0)	1 (2.9)	0 (0.0)	5 (13.5)	(6.3)
Mental disorder (*n* = 12)	8 (7.8)	1 (14.3)	2 (5.7)	0 (0.0)	1 (2.7)	(6.3)
Heart disease (*n* = 11)	1 (1.0)	0 (0.0)	1 (2.9)	0 (0.0)	9 (24.3)	(5.8)
Brain vascular disease (*n* = 9)	0 (0.0)	1 (14.3)	0 (0.0)	0 (0.0)	8 (21.6)	(4.7)
Chronic kidney disease (*n* = 5)	0 (0.0)	1 (14.3)	1 (2.9)	0 (0.0)	3 (8.1)	(2.6)
Other (*n* = 24)	11 (10.8)	0 (0.0)	3 (8.6)	1 (10.0)	9 (24.3)	(12.6)

EOAD: early-onset Alzheimer’s disease. MD: mixed dementia. FTD: frontotemporal dementia. PSP: progressive supranuclear palsy, VaD: vascular dementia. Values are expressed as *n* (%). Percentages are calculated within each diagnostic group (columns). The “Total” column indicates the absolute number and proportion of patients with each comorbidity in the overall cohort (*n* = 191). Because patients may present more than one comorbidity, totals do not sum up to 100%.

**Table 3 biomedicines-13-02793-t003:** Initial symptoms according to the type of YOD (*n* = 191) (Hospital Departamental Psiquiátrico Universitario del Valle during the period 2009–2024).

Initial Symptoms	EOAD (*n* = 102) n (%)	MD (*n* = 7) *n* (%)	FTD (*n* = 35) *n* (%)	PSP (*n* = 10) *n* (%)	VaD (*n* = 37) *n* (%)	Global (*n* = 191, %)
Memory (*n* = 96)	70 (68.3)	3 (42.9)	9 (25.7)	2 (20.0)	12 (32.4)	(50.1)
Depression (*n* = 57)	29 (28.0)	0 (0.0)	11 (31.4)	1 (10.0)	16 (44.1)	(29.8)
Delusion (*n* = 45)	24 (23.5)	3 (42.9)	7 (20.0)	2 (20.0)	9 (24.3)	(23.5)
Anxiety (*n* = 42)	21 (20.8)	1 (14.3)	11 (31.4)	3 (30.0)	6 (16.2)	(22.1)
Sleep disorder (*n* = 34)	20 (19.6)	0 (0.0)	4 (11.4)	1 (10.0)	9 (24.3)	(17.8)
Irritability (*n* = 28)	17 (16.8)	1 (14.3)	3 (8.6)	0 (0.0)	7 (18.9)	(14.7)
Movement disorder (*n* = 23)	4 (4.0)	0 (0.0)	4 (11.4)	8 (80.0)	7 (18.9)	(12.1)
Apathy (*n* = 15)	5 (4.9)	1 (14.3)	4 (11.4)	2 (20.0)	3 (8.1)	(7.8)
Language disorder (*n* = 13)	8 (7.8)	0 (0.0)	2 (5.7)	1 (10.0)	2 (5.4)	(6.8)
Self-care (*n* = 9)	6 (5.9)	0 (0.0)	1 (2.9)	0 (0.0)	2 (5.4)	(4.7)
Appetite/eating (*n* = 4)	3 (2.9)	0 (0.0)	1 (2.9)	0 (0.0)	0 (0.0)	(2.1)

EOAD: early-onset Alzheimer’s disease. MD: mixed dementia. FTD: frontotemporal dementia. PSP: progressive supranuclear palsy, VaD: vascular dementia. Values are expressed as *n* (%). Percentages are calculated within each diagnostic group (columns). The “Total” column indicates the absolute number and proportion of patients with each initial symptoms in the overall cohort (*n* = 191). Because patients may present more than one symptom, totals do not sum up to 100%.

**Table 4 biomedicines-13-02793-t004:** Current symptoms according to the type of YOD (*n* = 191) (Hospital Departamental Psiquiátrico Universitario del Valle during the period 2009–2024).

Current Symptoms	EOAD (*n* = 102) *n* (%)	MD (*n* = 7) *n* (%)	FTD (*n* = 35) *n* (%)	PSP (*n* = 10) *n* (%)	VaD (*n* = 37) *n* (%)	Global (*n* = 191, %)
Memory (*n* = 181)	100 (98.0)	6 (85.7)	32 (91.4)	8 (80.0)	35 (94.6)	(94.7)
Instrumental activities (*n* = 177)	94 (92.0)	6 (85.7)	35 (100)	10 (100)	32 (86.5)	(92.6)
Orientation (*n* = 154)	86 (84.3)	6 (85.7)	27 (77.1)	8 (80.0)	27 (73.0)	(80.6)
Anxiety (*n* = 151)	76 (74.5)	4 (57.1)	33 (94.3)	10 (100)	28 (75.7)	(79.1)
Basic daily activities (*n* = 149)	77 (75.5)	4 (57.1)	34 (97.1)	9 (90.0)	25 (67.6)	(78.0)
Depression/dysphoria (*n* = 132)	62 (60.8)	1 (14.3)	30 (85.7)	10 (100)	29 (78.4)	(69.1)
Irritability/lability (*n* = 128)	59 (57.8)	5 (71.4)	27 (77.1)	10 (100)	27 (73.0)	(67.0)
Nighttime behaviors (*n* = 106)	48 (47.1)	4 (57.1)	30 (85.7)	6 (60.0)	18 (48.6)	(55.5)
Apathy/indifference (*n* = 105)	52 (51.0)	2 (28.6)	34 (97.1)	7 (70.0)	10 (27.0)	(55.0)
Agitation/aggression (*n* = 85)	43 (42.2)	2 (28.6)	20 (57.1)	6 (60.0)	14 (37.8)	(44.5)
Appetite/eating (*n* = 82)	36 (35.3)	2 (28.6)	23 (65.7)	5 (50.0)	16 (43.2)	(42.9)
Delusions (*n* = 79)	40 (39.2)	3 (42.9)	19 (54.3)	2 (20.0)	15 (40.5)	(41.3)
Hallucinations (*n* = 62)	26 (25.5)	5 (71.4)	15 (42.9)	2 (20.0)	14 (37.8)	(32.5)
Disinhibition (*n* = 61)	32 (31.4)	2 (28.6)	16 (45.7)	2 (20.0)	9 (24.3)	(31.9)
Motor disturbance (*n* = 56)	20 (19.6)	1 (14.3)	8 (22.9)	10 (100)	17 (45.9)	(29.3)
Elation/euphoria (*n* = 17)	9 (8.8)	0 (0.0)	4 (11.4)	1 (10.0)	3 (8.1)	(8.9)

EOAD: early-onset Alzheimer’s disease. MD: mixed dementia. FTD: frontotemporal dementia. PSP: progressive supranuclear palsy, VaD: vascular dementia. Values are expressed as *n* (%). Percentages are calculated within each diagnostic group (columns). The “Total” column indicates the absolute number and proportion of patients with each current symptoms in the overall cohort (*n* = 191). Because patients may present more than one symptom, totals do not sum up to 100%.

**Table 5 biomedicines-13-02793-t005:** Linear regression of time from the onset of symptoms to the diagnosis of patients with YOD (*n* = 191) (Hospital Departamental Psiquiátrico Universitario del Valle 2009–2024).

Predictors	Non Standardized Coefficients		Standardized Coefficients	t	Sig.	95% Confidence Interval for B		Collinearity Statistics	
	B	Dev. Error	Beta			Lower Limit	Upper Limit	Tolerance	VIF
(Constant)	1.609	0.255		6.301	0.0001 *	1.105	2.112		
Symptom onset/first consultation time	0.918	0.062	0.710	14.723	0.0001 *	0.795	1.042	0.985	1.015
Affective disorder	1.101	0.481	0.112	2.291	0.023	0.152	2.050	0.954	1.048
Anxiety disorder	1.389	0.540	0.126	2.574	0.011	0.324	2.454	0.964	1.038
Type 2 diabetes mellitus	2.604	0.612	0.229	4.251	0.0001*	1.395	3.812	0.794	1.260
High blood pressure	−0.776	0.393	−0.106	−1.972	0.050	−1.552	0.001	0.795	1.258
Irritability	−0.897	0.423	−0.103	−2.122	0.035	−1.731	−0.063	0.969	1.032

B: non-standardized coefficient; Beta: standardized coefficient; Sig.: *p*-value; 95% CI = 95% confidence interval for B; Tolerance and VIF: collinearity statistics. VIF: variance inflation factor. * *p* value < 0.0001.The presence of T2DM was significantly and positively associated with the time prior to a definitive diagnosis (B = 2.604; *p* < 0.001), whereas HBP exhibited a marginally significant negative association with this factor (B = −0.776; *p* = 0.050). These findings indicate that patients who have been diagnosed with T2DM take, on average, 2.6 years longer to obtain a final diagnosis, whereas those with HBP spend 0.77 years in this stage.

## Data Availability

The data presented in this study are not publicly available due to privacy and ethical restrictions.

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
