# Peer review of "Young-Onset Dementia: Clinical Findings and Factors That Delay Early Diagnosis—A Retrospective Observational Study"

_biomedicines, 2025, doi:10.3390/biomedicines13112793_

Round 1
Reviewer 1 Report
Comments and Suggestions for Authors
Thank-you for the interesting article. I have added comments below for your consideration:
- At points in the article you refer to the same as patients with a diagnosis of probable YOD and at other times refer to the same sample as being patients with a diagnosis of YOD. For clarity you may wish to use the same terminology throughout the paper.
- Line 46: Should this say ‘….with symptoms beginning before the age of 65’
- Lines 78 – 80: I would suggest reviewing how the sentence is written, as the first consultation would not have been at the onset of symptoms.
- Line 106-108: Is there a reason why the diagnosis of dementia was according to ICD-10? Earlier in the article it states that DMS-5 criteria were used. It may need to be clarified if indeed both systems were used and if so the reasons for this.
- Line 136: Do you mean that the primary caregiver was female?
- Line 139 – 143: You state that the possibility of dementia was not considered in 41.4% of cases. Is this because it was not recorded in clinical records or is there another reasoning for this statement? My understanding is that both the ICD and DSM criteria state that other possible causes of cognitive impairment should be treated first prior to a diagnosis of dementia being made. It is possible that this was the reason for the alternative diagnoses in the first instance. On a related note, MCI is often considered as a possible prodrome to dementia, which could suggest that dementia was considered but that the patients were not considered to meet the full criteria required for a diagnosis of dementia at the time of diagnosis.
- Lines 159-160: This is the first time in the full body of the article that you have referenced HBP, T2DM and TBI. It would be worth checking if these should be written in full at this point (I note they have been written in full with the acronyms in the abstract).
- Line 259-261: Given that the article is about a cohort with YOD, do you need to include ‘especially among people below 65 years of age’.
- Libe 272: This is the first time you have referenced a delay in onset between 5 months and 18 years. I am unclear if this statement is in relation to your cohort, or based on the literature available. If it is in relation to your sample, did you account for the impact of outliers in your statistical analyses?
- Line 280: I am assuming you mean ‘the main modifiable risk factors’. The same applies to line 288.
- Line 344: You have the type 2 diabetes mellitus in full rather than using the acronym.
- Line 367: As per the comment relating to Lines 139-143, it may not be that the cognitive deficits were attributed to a metabolic cause, but that this was being ruled out as a cause first as per DSM and ICD criteria.
- In the limitations section:
- Consideration should be given to whether other ‘treatable’ causes were addressed first to rule these out, prior to a diagnosis of dementia being made (as is required by DSM and ICD guidelines). This will by necessitation increase the time to final diagnosis of YOD.
- It is unlikely that you can be 100% sure dementia wasn’t considered (and just not noted). As noted previously, MCI can be considered a prodrome so dementia may have been considered in these cases but that full criteria was not met.
- The sample (by necessity) only includes those who were referred to the hospital department
- The sample (by necessity) only includes those who received a diagnosis of YOD. It would be useful to noted that there will be a number who are still ‘misdiagnosed’ in the system, or have not been coded correctly in routinely collected data.
- It may be worth noting some of the limitations of using routinely collected information more generally.
Author Response
Dear reviewer, thank you for your clear and useful recomendations. Pleas see the attachment with our answers.

Reviewer 2 Report
Comments and Suggestions for Authors
The authors present a retrospective observational study on a cohort of 191 patients with Young-Onset Dementia (YOD) from a single center in Cali, Colombia. The manuscript addresses a relevant and clinically important topic, as YOD is often misdiagnosed, leading to significant delays in appropriate management. The analysis is methodologically sound, employing a linear regression model to identify predictors of diagnostic delay. While the paper is promising, several major and minor revisions are required before it can be considered for publication.
- The paper overstates some conclusions given the design and sample (retrospective, single-centre, selected patients). Reframe language throughout to emphasize associations rather than causation.
- The high rate of initial misdiagnosis, where over 40% of patients were first treated for other conditions like affective or anxiety disorders, is a key finding. However, the conclusion that this stems from a "lack of expertise on the part of health teams" is a strong assertion not directly measured by the study. The manuscript would be improved by rephrasing this to suggest that the data indicate potential gaps in clinical awareness or challenges in differentiating YOD from primary psychiatric disorders in its early stages.
- The methods need much more detail and transparency. The inclusion/exclusion process is unclear: how many records were screened, how many excluded (and why), and how many patients lacked sufficient data? Provide a STROBE flow diagram and a STROBE checklist.
- The diagnostic algorithm needs clearer description. The manuscript lists several diagnostic criteria sets (DSM-5, NINCDS-ADRDA, NINDS-AIREN, international FTD criteria, Movement Disorder Society criteria), but it is not clear which criteria were applied in which years nor whether biomarker data (amyloid/tau CSF or PET) were used to support AD diagnoses. Clarify whether all patients had the same diagnostic work-up (MRI, neuropsych tests, 18F-FDG PET) and whether FDG-PET was used for all cases. Provide inter-rater reliability (kappa) or describe the adjudication procedure used by the two reviewing psychiatrists.
- Important potential confounders appear missing (education, socioeconomic status, urban/rural residence, number of prior health-care contacts, family history of dementia, use of psychotropic medications before diagnosis). Consider including these if available, or acknowledge them as limitations.
- Provide the name of the ethics committee, approval reference number and approval date. For retrospective record reviews, state whether informed consent was waived.
- The sample is restricted to patients who eventually reached a final diagnosis in a psychiatric hospital, raising the possibility of selection bias and limiting generalizability. Diagnostic ascertainment is insufficiently transparent, as multiple criteria sets are listed without specifying which were applied to which disorders or years. It is also unclear whether all patients had uniform access to neuroimaging or biomarker studies, and no inter-rater reliability or adjudication procedures are reported for the two reviewing psychiatrists. Because the data span 15 years, secular trends in diagnostic practices and awareness are likely; the analysis does not address this.
- The presentation of results is inconsistent. Some variables are reported as means and SDs, others as medians and IQR, without clear justification. Several tables are poorly formatted, with unclear denominators and possible errors in percentages. Confidence intervals and exact p-values should always be reported rather than rounded values such as p=0.000.
- There are also concerns regarding funding. The text claims no external funding, yet the acknowledgments state hospital financing.
Author Response

(The authors gave the same response as above.)

Reviewer 3 Report
Comments and Suggestions for Authors
This retrospective observational study provides information on a cohort of 191 young-onset dementia (YOD) patients, examined from 2009-2024 in a single hospital. The authors collected data regarding the initial diagnosis, the final diagnosis, initial clinical presentation, comorbidities, time from symptom onset to first consultation and diagnosis, as well as the effect of these factors in delays in consultation and diagnosis. This manuscript provides data on a large single-center cohort of YOD over the course of 15 years.
Main comments:
- The authors should elaborate on the study design. The term "retrospective, analytical, observational study" needs clarification.
- Based on the 2.2. "Population and sample" subsection, all diagnoses (both baseline and final) were determined by two psychiatrists based on the data contained in the medical records of the patients. The authors should explain why the initial diagnoses were not used. If they were actually used, this point should be clarified.
- The authors used recent diagnostic criteria, which were not published at the time of patient evaluation in may cases. Were these criteria applied retrospectively to all patients based on medical files?
- The authors refer to MRI and PET-CT scans which were routinely performed in all patients. Since this approach (including standard PET-CT in all patients with cognitive deterioration) is exceedingly rate, authors should include these MRI and PET-CT data.
- In the baseline evaluation, none of the 191 patients received a definite diagnosis (e.g. AD, PSP, bvFTD etc.). Instead all patients had an unspecified dementia diagnosis (n=112). This is counter-intuitive, considering that many patients fulfill criteria for a specific cognitive disorder at first evaluation. The authors should provide reasons for the lack of more specific diagnosis at baselien.
- Among initial diagnosis, all patients with an affective disorder are classified as "bipolar disorder", with no patient receiving a diagnosis of depression or mania. The authors should elaborate on this matter.
- Final diagnoses included AD, VD, PSP, FTD, MD. Among the 191 patients, no patient received a diagnosis of DLB (the second most common cause of neurodegenerative dementia), corticobasal syndrome, primary progressive aphasia, posterior cortical atrophy, normal pressure hydrocephalus etc. The authors should provide a reason for the absence of these diagnoses in a cohort of 191 patients.
- The manuscript would benefit from providing a rationale or hypothesis regarding the correlations of carious comorbidities with time to first consultation or time to final diagnosis.
- The manuscript would benefit from a review by a native English speaker.
Comments on the Quality of English Language
The manuscript would benefit from a review by a native English speaker.
Author Response

(The authors gave the same response as above.)

Reviewer 4 Report
Comments and Suggestions for Authors
The work is well-written, well-illustrated with tables. However, there is too much information in the tables. The information is difficult to understand.
Some sentences are too long and complicated. It is difficult to grasp the idea. Breaking them into shorter sentences would improve clarity.
The summary and conclusions could be more concise.
The conclusions are difficult to understand or do they match the results.
Author Response

(The authors gave the same response as above.)

Round 2
Reviewer 2 Report
Comments and Suggestions for Authors
The authors have addressed the queries raised. The manuscript may be accepted.
Author Response
Dear reviewer, thank you for your comments, and approval.
Reviewer 3 Report
Comments and Suggestions for Authors
The authors have responded to all comments in a systematic and thorough manner. However, these responses have resulted in significant differences in study design (e.g. in the initial manuscript PET-CT was performed in all patients, in the revised version only in cases of diagnostic uncertainty). Additionally, two sets of diagnostic criteria were used (initially the ICD-10 and then the established diagnostic criteria for each disease), which is highly problematic, since differences will arise when classifying a patient based on different criteria. The authors support that all patients at baseline were classified as undetermined dementia because only clinical data were available at the time. However all diagnostic criteria are primarily clinical, and a diagnosis of probable PSP, possible bvFTD and possible AD can be established based on these clinical data. The authors have explained the absence of DLB and PD patients by introducing a selection bias (these patients are managed in a different setting).
Author Response
Dear reviewer, thank you for your comments. We have tried to improve the quality of our paper following your recomendations. Please see the attached archive with our answers.

Round 3
Reviewer 3 Report
Comments and Suggestions for Authors
The authors have replied to the queries. However, some major methodological issues, regarding two sets of diagnostic criteria, lack of an initial clinical diagnosis of the dementing disorders and absence of major dementia disorders (e.g. DLB,, NPH) in the cohort remain.